# Tribological and Chemical–Physical Behavior of a Novel Palm Grease Blended with Zinc Oxide and Reduced Graphene Oxide Nano-Additives

**Mohamed G. A. Nassef** [1,2], **Belal G. Nassef** [2,3,*], **Hassan S. Hassan** [4,5], **Galal A. Nassef** [2], **Marwa Elkady** [6,7] **and Florian Pape** [3]

1 Industrial and Manufacturing Engineering Department, Egypt-Japan University of Science and Technology, New Borg El Arab City, Alexandria 21934, Egypt; mohamed.nassef@ejust.edu.eg
2 Production Engineering Department, Faculty of Engineering, Alexandria University, Alexandria 21544, Egypt; galalnassef@alexu.edu.eg
3 Institute of Machine Design and Tribology, Leibniz University of Hanover, 30167 Hannover, Germany; pape@imkt.uni-hannover.de
4 Environmental Engineering Department, Egypt-Japan University of Science and Technology, New Borg El Arab City, Alexandria 21934, Egypt; hassan.shokry@ejust.edu.eg
5 Electronic Materials Research Department, Advanced Technology and New Materials Research Institute, City of Scientific Research and Technological Applications (SRTA-City), Alexandria 21934, Egypt
6 Chemical and Petrochemical Engineering Department, Egypt-Japan University of Science and Technology (E-JUST), New Borg El-Arab City, Alexandria 21934, Egypt; marwa.elkady@ejust.edu.eg
7 Fabrication Technology Department, Advanced Technology and New Materials Research Institute (ATNMRI), City of Scientific Research and Technological Applications (SRTA-City), New Borg El-Arab City, Alexandria 21934, Egypt
* Correspondence: nassef@imkt.uni-hannover.de

**Abstract:** The role of industrial lubricants in machinery is to reduce friction and wear between moving components. Due to the United Nations' tendency to reduce dependency on fossil fuel, a general awareness is strongly driven towards developing more eco-friendly lubricants. Palm oil possesses promising properties, which promote it to be a competitive alternative to the hostile mineral oils. Still, marginal oxidation stability, viscosity, and tribological properties remain critical issues for performance improvement. This paper presents an improved palm grease using reduced graphene oxide (rGO) and zinc oxide (ZnO) nano-additives at different concentrations. Oil and grease samples were tested for viscosity, oxidation stability, pour point, penetration, roll stability, dropping point, churned grease-oil release, copper corrosion, friction, and wear. ZnO additives enhanced the oxidation stability by 60% and shifted the pour point to 6 °C. Adding ZnO and rGO to the palm grease increased the load-carrying capacity between 30% and 60%, respectively, and reduced the friction coefficient by up to 60%. From the wear scar morphologies, it is believed that graphene 2D nanoparticles formed absorption layers which contributed to the increase in load-carrying capacity, while ZnO chemically reacted with the metallic surface layer, forming zinc compounds that resulted in a protective boundary lubricating film.

**Keywords:** palm oil; grease synthesis; viscosity; oxidation stability; nanoparticles; coefficient of friction; wear resistance

## 1. Introduction

Current studies reveal that almost 30% of the overall world's energy consumption is directed towards the industrial sector, of which 40% results in dangerous carbon emissions [1,2]. According to the International Energy Agency (IEA) statistics, the industry sector contributed by around 9.0 Gt of $CO_2$ emissions in 2022, which is 25% of the total world's energy system emissions [3]. These emissions are unsatisfactory and should be

reduced by 25% by 2030, targeting net zero emissions in 2050 to keep global warming below 2 °C [4–6]. Another challenge that worsens the situation is the rapid increase in energy demand from manufacturing industry and related logistic industries [7]. This highlights the critical need for the development of more energy-efficient systems. For example, Denkena et al. [8] summarized different recent measures to achieve efficient energy systems and operations to reduce ecological impact as well as energy demand. Furthermore, the replacement of conventional lubricants with more energy-efficient ones has been found to be an appealing solution to reduce the carbon footprint of machinery [9].

Despite the rapid increase in the lubricant market share [8], the emitted byproducts from conventional mineral oil lubricants still do not conform with the environmental preservation and sustainability regulations [10,11]. It was noted that about 55% of petroleum-based lubricants are annually harming the environmental resources represented in water, air, and soil [12,13]. Considerable research attention is devoted to mitigating these environmental threats of mineral lubricants represented in the high toxicity and low biodegradability by their replacement with more eco-friendly lubricants [12–15].

Biolubricants are deemed to be a promising solution to lessen the emitted greenhouse gases from utilizing petroleum-based lubricants [14]. They are biodegradable hydrocarbons, which are extracted from biomaterials found in animal fats, plant oil, and microalgae [16–19]. Biolubricants offer remarkable properties such as high lubricity, low volatility, satisfactory viscosity index, and high flash point [20]. A special interest has been directed towards vegetable oil due to its superior biodegradability over animal fats and esters [18]. Moreover, vegetable oil absorbs considerable amounts of $CO_2$ through the photosynthesis process as compared to that released due to the burning process [21].

Many researchers investigated the tribological and chemical–physical properties of vegetable oils to replace mineral ones in industrial applications [22–25]. For example, experimental investigation tested 12 different plant oils for tribological properties and showed that some vegetable oils can form thicker lubricating films than that predicted from elastohydrodynamic theory [26]. Another recent finding proved that vegetable oil-based grease produces a stable and consistent hydrodynamic film lubricant with higher load-carrying capacity in comparison with commercial lithium grease [27,28].

Palm oil is one of the most commonly studied bio-oils due to its highest production-rate-to-price ratio (4 MT/ha) compared to other vegetable oils [29]. It contains free fatty acids, which are in the form of linear carboxylic acids of carbon atoms ranging from 12% to 24%. Oleic (monosaturated), linoleic (polyunsaturated), palmitic, and stearic acids are the main components of the fatty acid chain [30]. These acids are responsible for the distinguished oil's performance in reducing wear and friction [31]. Aiman et al. [32] reported a 45% reduction in the coefficient of friction (COF) after replacing the engine oil with refined, bleached, and deodorized (RBD) palm olein tested under higher loading conditions. Other studies in [33,34] investigated the tribological behavior of RBD palm olein mixed at different blends with mineral oil for reciprocating machinery. In [33], it was found that the viscosity index (VI) and flash point gradually increased until reaching maximum values at a mixture of 60% mineral oil and 40% RBD. In this case, the friction torque and wear scar diameter (WSD) were reduced by 31% and 33%, respectively.

Despite the promising findings from previous studies, palm oil still suffers from drawbacks such as poor oxidation stability and marginal viscosity values, which limits its application in industry. Therefore, investigators conducted many studies addressing these issues by testing additives on palm oil lubricants [35–38]. Traditional additives such as zinc dialkyldithiophosphates (ZDDP) have a negative impact on the environment, which involves high contents of toxic sulfur and phosphorous in the blend, the corrosive metallic compounds (such as lead), and tacky solubility of additives in oil [39,40].

Nanomaterials [41], such as activated carbon nanoparticles [42], boron nitride, carbon nanotubes, graphene, and its oxide [43,44], are considered as more eco-friendly anti-wear (AW) and extreme pressure (EP) additives to lubricants. Furthermore, toxic func-

tional groups in synthetic oils were found to be effectively removed using biochar as an antioxidant agent [45]. Effective AW agents represented in metal oxide nano-additives, such as titanium dioxide [46,47]. ZnO [48–51] acts as an antioxidant and AW agent that stabilizes the physical properties of oil under harsh conditions.

Bhaumik et al. [48] conducted a detailed study of the effect of adding different weight percentages of ZnO to castor oil on its tribological behavior. They found that the addition of 0.1 wt.% ZnO reduces the COF and WSD by about 25% and 44%, respectively, as compared to the base oil. The effect of mixing 0.01 wt.% multilayer graphene with a wax separated from agro-waste resources on the properties of the biogrease itself was studied under different loading and temperature conditions [52]. Graphene was found to act as an efficient lubricant additive by decreasing the wear rate by about 35%.

Roselina et al. [53] conducted experimental work to enhance the viscosity index (VI) of both palm and synthetic (SAE 0W20) oils through adding $TiO_2$ nanoparticles. The attempt showed that VI recorded an increase of up to 4% and 7% for palm and synthetic oils, respectively. Tan et al. [54] evaluated the oxidation stability of palm oil after adding 0.5 wt.% zeolite nanoparticles. They found that zeolite nano-additives decelerate the oxidation process of palm oil through the good extraction and adsorption of hydroperoxides preventing propagation and subsequent cleavage of C=C [55,56].

Although nano-additives proved to enhance the biolubricant characteristics, the toxicity effect of the applied nanoparticles should be considered to avoid damaging the biodegradability of the lubricant itself. Therefore, previous studies focused on ZnO and $TiO_2$ due to their antioxidant characteristics along with their lower toxicity [57]. Other investigations [58–61] recommended graphene due to its distinguished 2D multilayered nanostructure. Reduced graphene oxide offers outstanding tribological and physical properties, as well as its green characteristics represented in controllable function groups with the absence of harmful elements such as sulfur and phosphorous [62].

From the review of the literature, insufficient works exploited the advances in metallic and carbonaceous nano-additive materials for the development and enhancement of palm grease as a green alternative to commercial grease. On the other hand, attempts made so far to develop palm grease without additives or modifications have shown unsatisfactory results in terms of very low NLGI grade [63]. The trials to apply some additives such as fumed silica did not enhance palm grease corrosive resistance or COF [64]. Furthermore, modification of palm grease by esterification would increase jeopardize its economic and ecofriendly advantages [31]. This study presents a novel synthesized palm grease lubricant reinforced with different blends of rGO and ZnO nano-additives aiming to enhance its chemical–physical and tribological performance. In previous works by the authors, rGO was applied to commercial lithium grease at 0.5, 1, 2, 3.5, and 5 wt.% blends, and only the first three blends were found to be effective in power reduction of rolling bearings, while the blends with rGO higher than 2 wt.% showed undesired agglomeration. Therefore, 0.5, 1, and 2 wt.% were selected for this investigation. Similarly, this work tested ZnO at a three concentration values based on previous findings [50,51,65]. Kinematic viscosity, pour point, and oxidation stability tests were conducted on palm oil samples with and without nano-additives. Another set of tests were carried out on grease samples for evaluating grease consistency, roll stability, dropping point, churned grease-oil release (CGOR), and copper corrosion. Brugger customized test setup and 4-ball wear test were used to determine wear scar morphology, COF, and load-carrying capacity for the synthesized palm grease samples. Results are compared with lithium grease sample.

## 2. Materials and Methods

### 2.1. Nanomaterials Characterization

ZnO nano powder is prepared using the sol–gel technique by dropping ammonium hydroxide solution (1 M) into 10 mL of 1 M zinc acetate aqueous solution with constant

stirring, pH adjustment at 10, and constant temperature of 80 °C overnight. The obtained white fine powder is then filtered and washed several times using ethanol and distilled water to remove any residual salts. Finally, the washed precipitate is centrifuged for 30 min at 4000 rpm and then dried at 70 °C, whereas reduced graphene oxide (rGO) is purchased from the local market in the form of nanosheets. The graphene is manufactured by the modified Hummer's method, in which thiourea is used as a reduction agent to reduce graphene oxide into reduced graphene oxide, based on which N and S are dopants, which are revealed from thiourea structure.

To investigate the structural characteristics of rGO and ZnO nanoparticles, in terms of surface morphology, particle size, particle shape, and the extent of particles aggregation, a scanning electron microscope (SEM) (JEOL JSM-IT200, JEOL Ltd., Tokyo, Japan) and a transmission electron microscope (TEM) (JEOL JEM-2100, JOEL Ltd., Tokyo, Japan) are used. The preparation procedure of SEM samples involves assigning the powder sample to a double-face carbon tape after being coated with platinum–palladium at 40 mA using JEOL, JEC-3000 FC Auto Fine Coater. The TEM sample preparation relies mainly on the dispersion and sonication of nanoparticles in ethanol solution for 15 min. Then, a tiny droplet of the solution is applied to a copper–carbon grid through the drop-casting technique, tested under 200 kV.

Furthermore, the identification of microstructural phases and crystallinity percentage is performed by studying the characteristic peaks of each nanomaterial using X-ray diffraction (XRD) spectroscopy (Empyrean Malvern Panalytical, Almelo City, The Netherlands). XRD spectra are collected at 40 keV and 30 mA using Cu-K$\alpha$1 radiation. Regarding the quality of the nanostructure, the function groups attached to the structure of both rGO and ZnO nanomaterials are studied using Bruker Vertex 70 Fourier-transform infrared (FTIR) spectroscopy (Bruker Company, Billerica, MA, USA) with a detection range of 400–4000 cm$^{-1}$.

Finally, detailed information about rGO structural defects, along with its number of layers, is obtained using Raman spectroscopy (WITec alpha 300 R, Ulm, Germany). The test starts with exposing the rGO sample to 3 mW power under an applied laser line of 532 nm wavelength. Then, the final spectrum is captured and displayed in a 20 s detection time with five accumulations.

*2.2. Palm Grease Synthesis*

Palm grease is synthesized in this work using 80% of palm olein as a green base-oil and 20% of 12-lithium hydroxystearate as a thickener to compose the final grease structure, which is prepared through three main steps [24,53]. The first step involves adding 70% olein in a 2 L measuring flask. The flask is heated using an MSH-20D magnetic stirrer till reaching 90 °C. After holding for 10 min, the soap is gradually added to the flask with a total weight percentage of 20%. Afterward, the temperature is increased to 250 °C with simultaneous mixing for 1 h in order to ensure the soap dissolution. Secondly, the mixture is cooled down to 120 °C, at which the remaining 10% of the olein is added and mixed with the total blend. Finally, the obtained blend is homogenized for 30 min at room temperature using a commercial mixer to ensure a suitable dispersion of thickener in the base olein. The obtained grease sample is characterized through FTIR and compared to the plain oil FTIR to confirm the transition from oily state to the grease state. Palm grease samples with different concentrations of ZnO and rGO nano-additives are prepared and labeled for chemical–physical and tribological tests, as shown in Table 1. Samples of commercial lithium grease with 80% mineral oil and 20% thickener are also tested under the same conditions for comparison.

**Table 1.** Test grease samples with different concentrations of ZnO and rGO nano-additives.

| Grease Sample | Grease Blends |
| --- | --- |
| Palm Grease A | Palm-oil-based lithium grease (without nano-additives) |
| Palm Grease B | Palm grease with 0.05 wt.% ZnO |
| Palm Grease C | Palm grease with 0.1 wt.% ZnO |
| Palm Grease D | Palm grease with 0.25 wt.% ZnO |
| Palm Grease E | Palm grease with 0.5 wt.% ZnO |
| Palm Grease F | Palm grease with 0.5 wt.% rGO |
| Palm Grease G | Palm grease with 1 wt.% rGO |
| Palm Grease H | Palm grease with 2 wt.% rGO |

### 2.3. Palm Grease Consistency Test

The consistency of the prepared grease is an important measure of grease rigidity; hence, it dictates grease suitability for bearings lubrication. It is evaluated according to the resistance of the grease–fibrous structure to a known dropping load. Adequate consistency of palm grease ensures its firmness and provides stable channels of oil. These channels act as reservoirs through which the mating surfaces interact with the oil during service time.

Hence, a cone penetration test (ASTM D217) is used to obtain the relative stiffness of the palm grease following the provided National Lubricating Grease Institute (NLGI) scale. The testing procedure is conducted in two cases: unworked penetration (no preconditioning of the grease) and worked penetration. The purpose of the worked penetration case is to mechanically stress the palm grease using a plunger, making the new grease consistency more comparable to that of the grease in service.

In the unworked penetration test, the palm grease sample of 500 g is placed in a suitable worker cup at 25 ± 5 °C, after which the penetrometer is freely dropped into the sample to penetrate it for 5 s. The worked penetration test relies on shearing the prepared grease before penetrating it through subjecting the sample to 60 double strokes. NLGI grade was obtained by measuring the penetration depth in a tenth of millimeters.

### 2.4. Roll Stability

The mechanical stability of grease is another essential characteristic that determines to what extent the grease would maintain its consistency during mechanical shearing in critical machinery components, such as rolling bearings. A roll stability test (ASTM D1831) evaluates the mechanical stability of palm grease in terms of the observed change in the readings of the cone penetration test before and after the mechanical action. The test starts with the transfer of a grease sample of 5 g to a rotating steel cylinder in the test equipment. Thereafter, a 5 kg roller is placed into the cylinder, which is left to rotate at 170 rpm for about 2 h. Finally, the grease shear stability is calculated by subtracting the two penetration values.

### 2.5. Churned Grease-Oil Release (CGOR) Test

Another key factor that governs the overall grease performance in rotating machinery components such as rolling bearings is the oil separation amount from the thickener. During operation, two phases of lubrication take place: grease churning and oil bleeding. In the churning phase, grease is pushed away by the running action of rolling elements toward the unswept areas (bearing cages, bearing shoulders, or seals). Then, in the bleeding phase, the formed grease on the sideways bleeds oil to the contacts driven by capillary forces, surface tension, and centrifugal forces. The amount of oil separation is controlled by the thickener type, amount, and grease preparation process [66].

### 2.6. Copper Corrosion Test

The copper corrosion strip test measures the relative degree of corrosivity of petroleum products. The test is performed according to ASTM D4048 to confirm that the lubricating grease has no harmful sulfur elements that can react with the copper coupon sample and tarnish it. The test is an acceptance test of palm grease as it checks its ability to protect the mating surfaces of rotating machinery components during service time. A polished copper strip sample is completely immersed into a tightly packed sample of grease, which is heated in an oven or liquid bath at a specified temperature for a defined period of time. The test conditions are 100 °C +/− 1 °C for 24 h +/− 5 min. By the end of the heating period, the copper sample is removed from the grease sample and washed with acetone. Finally, a visual inspection is made by comparing the copper strip to the ASTM Copper Strip Corrosion Standards.

### 2.7. Dropping Point Test

In order to check the heat resistance property of grease, the dropping point of prepared grease samples is calculated according to ASTM D2265. A grease sample of 10 g is placed into an aluminum block oven using a cup, which is made of chromium-plated brass and supported by a glassy test tube. The oven temperature is then increased to reach 232 °C and monitored by an attached thermometer. After reaching the dropping point, an oil drop falls from the cup to the test tube. At this point, the thermometer records the sample temperature, which is used to calculate the grease dropping point.

### 2.8. Rheological Test

The rheological characteristics of palm grease are evaluated in comparison to a commercial lithium grease using a rheometer (Kinexus Prime lab +, Netzsch GmbH, Selb, Germany) following DIN 51810-1, as shown in Figure 1. Tests are conducted at temperatures of 20, 40, and 60 °C to assess non-Newtonian behavior and thermal response. The test involves applying a shear rate range of 0.1 to 100 s$^{-1}$ to a 1.7 mL volume of the grease sample. To ensure reliability, the test is repeated three times for repeatability. The results are analyzed using a power law model to characterize the non-Newtonian behavior of the greases and to evaluate their thermal behavior as temperature increased.

Furthermore, the viscosity of the plain oil is assessed with and without nano-additives in accordance with ASTM-D2196-20. This test is mainly dedicated to obtaining the appropriate kinematic viscosity value, which is crucial to specify the optimum operating condition for the used mixtures. An oily sample of 0.3 mL is applied to the rheometer lower plate followed by allowing a conical plate of the same material to make contact with the sample at 40 °C, then 100 °C, while applying the same shear rate range as in the grease sample.

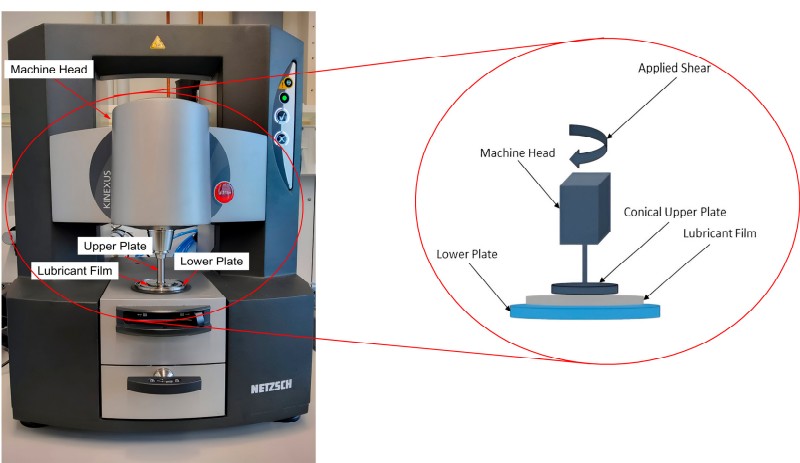

**Figure 1.** Rheometer setup.

*2.9. Pour Point Test*

The direct way to have a significant indication of the fluidity of palm oil is to test its pour point, which is defined as the lowest temperature that the oil can reach without being solidified. Pour point is a key property of lubricant quality as it is important to ensure adequate lubrication of machinery operating in low-temperature regions. This property is tested according to ASTM-D97-17b by pouring an oily sample of 30 mL into a test jar, which is exposed to sufficient heat till 45 °C. Then, the jar is left in a cold bath to cool down, and the pour point value is measured using a higher-range thermometer. The pour point is 3 °C higher than the temperature at which the oil no longer flows under gravity in a horizontally fixed sample container for 5 s.

*2.10. Oxidation Stability Test*

Oxidation stability plays a vital role in determining the grease service life while being operated in machinery based on the time required to oxidize the base oil, according to ASTM D-2272 for steam turbine oils. The test begins by inserting a 50 g sample of palm olein (base oil without and also with additives) samples in a glass container along with water and copper catalyst. Then, the container is placed inside a rotating oxygen-pressurized vessel, which rotates at 100 rpm, making a 30° angle with the horizontal line. The pressure and temperature inside the vessel account for 620 kPa and 150 °C, respectively. Finally, the oxidation stability is evaluated by calculating the oxidation time, which is recorded by measuring the time till the pressure value plummets to 175 kPa [67].

*2.11. Palm Grease Tribological Tests*

The load-carrying capacity of each grease blend is determined using a tailored test setup based on Brugger's test, as shown in Figure 2a. Its principle is based on the roller-on-ring test according to DIN 51347-1. During each test, the grease sample (8 g) is placed between a roller element of 15 mm width and 11 mm diameter and a 25 mm diameter ring which is set to rotate at 800 ± 5 rpm by an electric motor. The roller and the ring axes are set at 90° apart. The roller is located at one side of a compound lever mechanism, and on the other side, a hinge is used for attaching test loads of 500 g each.

The electric motor drives the ring for 30 s to allow the grease sample to generate a consistent lubricant film between the ring and roller element. Afterwards, a step load is attached to the hinge of the lever mechanism for 10 min ± 15 s. Then, the motor is turned off, and the roller element is removed to examine the worn surface. A new test roller is attached to the lever mechanism with additional load step added to the hinge. The steps of each test run are repeatedly carried out, and in each trial, another load step is added until welding takes place, which causes seizure of the motor. This phase indicates a complete breakdown of the grease film separating the roller element and ring. The test is repeated three times to ensure the obtained results, and the average limiting load is calculated. The load-carrying capacity is then identified as the maximum load step at which the grease film kept the two rolling elements at separation and above which seizure occurs.

The tribological properties of palm grease samples are evaluated in terms of COF using four-ball wear tests according to ASTM 5183 as shown in Figure 2b. First, the cup is filled with grease blend and the cup temperature is adjusted to 75 ± 2 °C. Afterwards, the upper ball inside the cup is pressed against three lower balls using a static load of 40 kg for 60 min ± 1 min. The device actuator drives the upper ball at a rotational speed of 600 ± 60 rpm, which ensures a continuous sliding mode operation under boundary lubrication condition. To calculate the COF for each grease sample, an incremental weight of 10 kg is added every 10 min until reaching the seizure condition, according to ASTM 5183. The tests are repeated three times for each grease blend to calculate the average COF value.

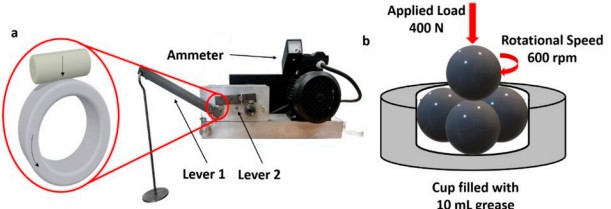

**Figure 2.** (**a**) A tribotester based on Brugger's test. (**b**) Principle of four-ball tribological test [43].

## 3. Results and Discussion

### 3.1. Characterization Results

Figure 3 shows the SEM and TEM micrographs for ZnO nanoparticles. SEM results are obtained at a magnification of 20,000× and a resolution of 1 μm, revealing the irregular structure of ZnO with a rosette shape of crystals and a wurtzite hexagonal phase, which was confirmed by previous investigations [68,69]. The previous findings from the SEM micrograph are verified by the TEM test, which reveals the hexagonal nanoparticles with a range of size between 26–64 nm and an average size of around 45 nm.

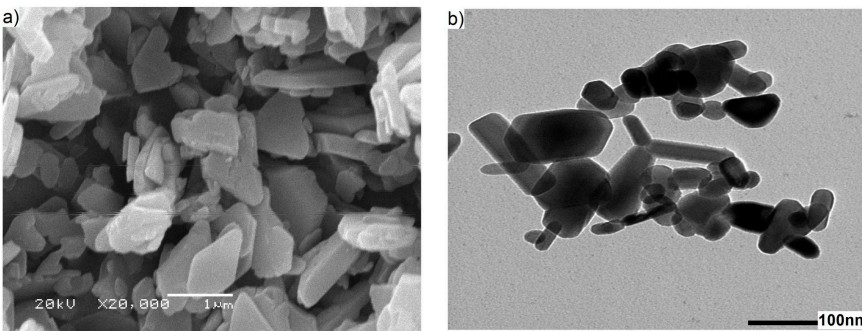

**Figure 3.** (**a**) SEM micrograph. (**b**) TEM micrograph of ZnO.

The XRD graph of ZnO nanoparticles is represented in Figure 4a. The diffraction pattern consists of a series of angles (2θ) lying at 31.6, 34.28, 36.08, 47.4, 56.42, 62.74, 66.18, 67.78, 68.96, and 76.98, which are assigned to the hkl values of (100), (002), (101), (102), (110), (103), (200), (112), (201), and (202), respectively [49]. These peaks provide an indication of a highly pristine ZnO structure due to the lower peaks of CH and CH2, as well as the intermediate presence of C-O. The peaks confirm that the ZnO material exhibits a clean and well-defined composition [70]. The strong presence of the (101) peak and its very sharp feature demonstrates the well-configured crystalline structure of the nanoparticles, which coincides with the JCPDS card no. 36-1451 [71]. According to the Debye–Scherrer formula, the crystallite size is estimated at 30.10 nm at full width half maximum (FWHM) value of 0.29.

Figure 4b displays the FTIR spectral analysis of ZnO nanoparticles, revealing the attached functional groups and their bonding type. It is found that the sample has an intermediate adsorption peak at 557.98 cm$^{-1}$, which refers to a metal oxide (Zn-O) function group. Also, this peak ensures the stable and well-established ZnO structure, confirming the XRD results, which have been reported by previous investigations [72,73], while the revealed vibrational peaks at a wavenumber range starting from 800 to 1000 cm$^{-1}$ are associated with the Zn-OH group [74]. Moreover, the hydroxyl (O-H) stretching bands are revealed at both wavenumbers of 3441.30 and 2923.90 cm$^{-1}$, indicating the water contained in the ZnO structure [75]. The C-O function group shows a medium presence in the spectrum at 1633.42 cm$^{-1}$, which may be due to the introduced carbonaceous compounds during the fabrication of ZnO [76]. However, the vibrational peaks that appear at 1260.84 and 1383.22 cm$^{-1}$ belonging to C-H$_2$ and C-H bands, respectively, refer to the monoacetate group represented as an intermediate compound [77].

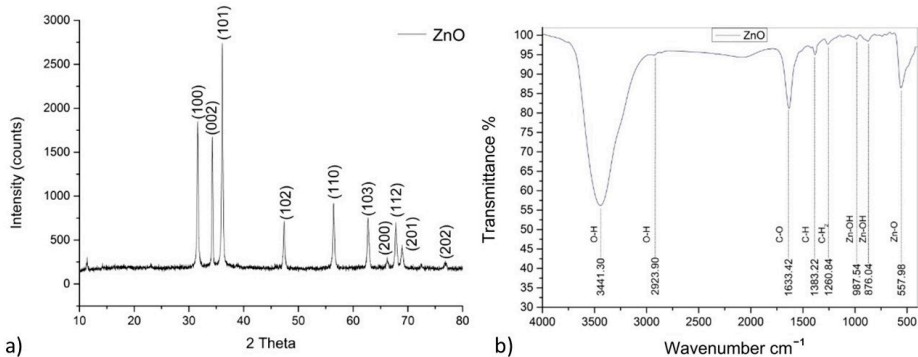

**Figure 4.** (**a**) XRD pattern of ZnO nanoparticles. (**b**) FTIR spectrum of ZnO nanoparticles.

Figure 5 represents SEM and TEM micrographs of the rGO nanostructure. By studying the SEM images, graphene is found to have a less-wrinkled nanosheet morphology with an aspect ratio (length divided by thickness) of about 6. The previous findings are confirmed by TEM, which shows a layered and transparent structure of graphene consisting of four less folded and stacked layers. Regarding the XRD analysis (shown in Figure 6a), a remarkable near-sharp characteristic peak appeared at $2\theta = 24.63°$, which is attributed to the basal plane (002). The presence of this characteristic peak confirms the appropriate reduction of graphene oxide (GO) during the graphene synthesis process. Also, the near-sharp feature ensures the few-layered graphene structure with a crystallinity percentage of 54%. Furthermore, the d-spacing (interlayer distance) and the average grain size (at full width half maximum (FWHM) of 5.89) were estimated at 0.37 nm and 1.44 nm according to Scherrer's and Bragg's rules, respectively.

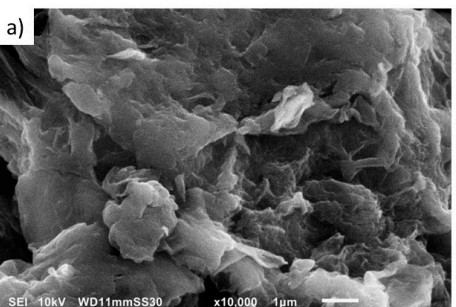
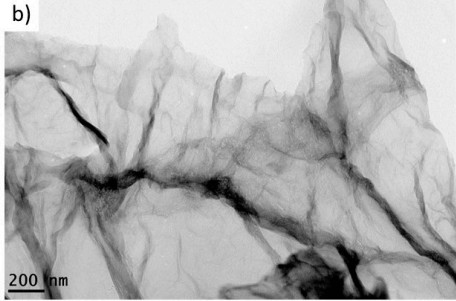

**Figure 5.** (**a**) SEM micrograph. (**b**) TEM micrograph of rGO.

Figure 6b shows the FTIR spectrum of rGO at a wide range of wavelengths, starting from 500 to 3500 cm$^{-1}$. The significant vibrational peaks were observed at 3421.43, 1629.73, 1386.57, 1191.48, and 754.28 cm$^{-1}$. It was found that the O-H function group exhibited a strong presence at a wavelength of 3421.43 cm$^{-1}$, referring to the amount of water that existed in the graphene structure, while a reasonable degree of structural integrity was inferred from the C=C stretching band, which showed a medium peak at 1629.73 cm$^{-1}$. Another weak vibrational peak was revealed at 1191.48 cm$^{-1}$ (C-O), indicating the insufficient amount of oxygen species needed to penetrate the covalent basal plane bonding, which emphasizes the chemical reduction of GO. Furthermore, the wavenumbers of 1386.57 and 754.28 cm$^{-1}$ are attributed to C-H bending, which showed a negligible presence due to the scarce functionalization of hydrogen species with carbon atoms. The presence of the N=C=S functional group reveals possible but insignificant contamination during sample preparation or impurities in precursor materials during the synthesis process.

Turning to the Raman results (shown in Figure 6c), three characteristic peaks were observed at 1350, 1595, and 2925 cm$^{-1}$, expressing D, G, and 2D bands, respectively. Regarding the D-band, it is mainly responsible for structural deficiencies. However, the

G-band identifies the ordering and symmetry of graphene layers. As for the 2D-band, it symbolizes the stacking of layers through the whole graphene structure. Concerning the structural disorders, the D-band showed a remarkable appearance in the spectrum, which refers to the significant amount of edge defects induced during the reduction of C=C compounds. The previous finding was supported by the calculated intensity ratio of ID/IG, which accounted for 0.98. The I2D/IG intensity ratio was found to be 0.91, which confirms the TEM findings about the number of graphene layers.

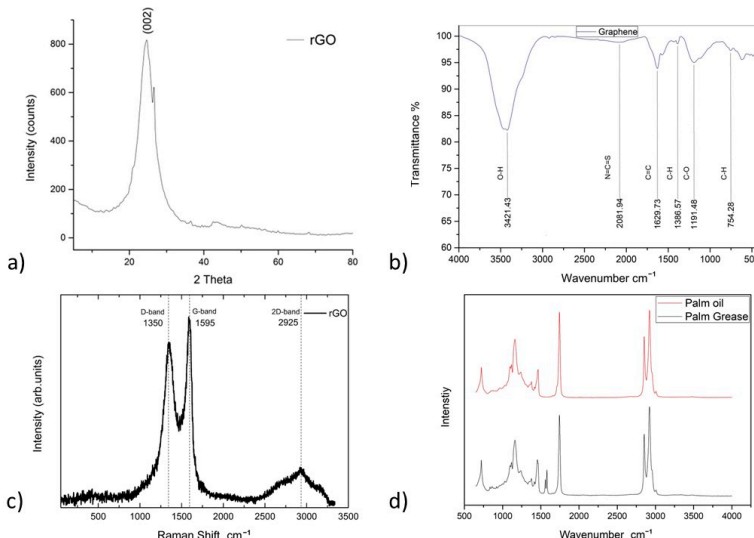

**Figure 6.** (**a**) XRD pattern, (**b**) FTIR analysis, (**c**) Raman spectrum of rGO, (**d**) FTIR of palm grease vs. palm oil.

The FTIR spectrum of the prepared grease, compared to the structure of palm oil, shows similar functional groups with notable differences, as shown in Figure 6d. Both spectra exhibit nearly identical functional groups, except for the prominent carboxylate group observed at 1582 cm⁻¹, corresponding to stretching vibrations of the carboxylate ion (COO⁻). This peak is completely absent in the palm oil spectrum. Also, a stronger presence of C-H asymmetric stretching band was observed as compared to palm oil. This matches with the finding of a previous research article which confirms Li-based grease structure [78].

### 3.2. Chemical–Physical Characterization Results

Table 2 shows the chemical–physical properties of palm grease and commercial lithium grease with 80% mineral oil and 20% lithium thickener. The palm grease has a homogenous creamy beige colored appearance. The obtained penetration levels are found to be around 216 mm for unworked conditions. The value was slightly increased after 60 strokes to 200 mm, which reveals shear thinning behavior of palm grease. This is due to the elastic deformation of the 3D fibrous structure of the thickener and reduction of base oil viscosity under shear loads. The penetration results reveal that palm grease rigidity can be classified as NLGI grade 3, which is higher than lithium grease grade (NLGI 2). The amount of lithium thickener is the key parameter in reaching adequate NLGI grade, as it acts as a resilient reservoir with its 3D fibrous structure network that releases the base oil under force application. Since palm grease has the same amount of lithium thickener as commercial grease, the observed difference in grease consistency is related to the preparation process and the base oil viscosity index, which is a controlling parameter in defining the grease rigidity.

The penetration values are also confirmed by the shear stability results of palm and lithium greases. From the roll stability test, the change in penetration levels for palm and lithium grease samples were found to be equal to 16 and 15, respectively. Hence, the

post-penetration levels are higher than prepenetration values, which confirms the shear thinning behavior of the grease structure. The change in penetration levels is found to be at a reasonable accepted value according to ASTM D1831. It was expected that palm grease would have the same mechanical stability as lithium grease, as both have the same thickener; however, palm oil contains stearic acid, which results in a more resilient fibrous structure in the form of long entangled and twisted ribbons [79].

During the operation of a machinery component such as rolling bearings, the grease lubrication exhibits two phases: churning phase and bleeding phase. In the churning period, the grease is swept away from the raceways and the thickener microstructure is subjected to thermomechanical degradation by drag forces and shearing action. This leads to aging of the thickener fibrous structure and reduction of the consistency of the churned grease. In return, the oil bleeding rate will increase, indicated by undesired high oil separation mass %. From the CGOR test, the oil separation of palm grease experienced a maximum of 8.63% when tested at 40 °C, suggesting good colloidal stability, conforming with the shear stability results.

The good consistency of palm grease Is reflected in the copper corrosion test results. Palm grease showed an insignificant degree of copper corrosion (1a), which is similar to lithium grease. The slight tarnish of copper coupon samples for palm grease samples with and without nano-additives reveals good protection for surfaces against chemical reactions and confirms the absence of harmful sulfur element. On the other hand, the dropping point of palm grease reached 209 °C, a value that indicates higher thermal resistance of palm grease (by 32%) than lithium grease (160 °C). Beyond this dropping point, the bonds holding the thickener compounds of lithium and stearate begin to break down, allowing the oil to be separated easily. Hence, the higher value of the dropping point refers to a stronger cohesiveness of palm grease structure [80], which conforms with the penetration and oil separation results.

**Table 2.** Physicochemical results of test grease samples.

| Test Description | Test Standard | Commercial Lithium Grease | Palm Grease A | Palm Grease B | Palm Grease C | Palm Grease D | Palm Grease E | Palm Grease F | Palm Grease G | Palm Grease H |
|---|---|---|---|---|---|---|---|---|---|---|
| Penetration, 60× (0.1 mm) | ASTM D217 | 255 ± 5 | 200 ± 8 | - | - | - | - | - | - | - |
| Penetration, unworked | ASTM D217 | 270 NLGI 2 ± 7 | 216 NLGI 3 ± 9 | - | - | - | - | - | - | - |
| Roll stability, penetration change (0.1 mm) | ASTM D1831 | −15 | −16 | - | - | - | - | - | - | - |
| Dropping point (°C) | ASTM D2265 | 160 ± 2 | 209 ± 6 | - | - | - | - | - | - | - |
| Color | Visual | whitish | Beige | - | - | - | - | - | - | - |
| Churned grease oil-release, oil separation (mass%) | AMS 1066 | 1.6 | 8.63 | - | - | - | - | - | - | - |
| Kinematic viscosity at 40 °C (cSt) | ASTM D445 | 45 | 47.5 | - | - | - | - | - | - | - |
| Kinematic | ASTM | 4.5 | 6.39 | 7.82 | 8.68 | 7.98 | 8.54 | 8.20 | 8.30 | 8.70 |

| viscosity at 100 °C (cSt) | D445 | | | | | | | | | |
|---|---|---|---|---|---|---|---|---|---|---|
| Copper corrosion at 100 °C, 24 h | ASTM D4048 | 1A | 1A | 1A | 1A | 1A | 1A | 1A | 1A | 1A |
| Oxidation stability of base oil | ASTM D2272 IP 229 | 27 min ref [67] | 30 min | 32 min | 35 min | 40 min | 48 min | - | - | - |
| Pour point (°C) | ASTM D7346 | | 6 | 6 | 6 | 6 | 6 | 9 | 9 | 9 |

Figure 7 illustrates the rheological behavior of both palm grease and lithium grease. Both greases exhibit shear-thinning behavior, which is typical for lubricating greases used in machinery. However, the palm grease sample demonstrates higher viscosity values at 20 °C, primarily due to the partial solidification of palm oil at this temperature. These results increase the resistance to shear motion within the contact surfaces. As the temperature increases, the viscosity values decrease significantly, with the lowest values observed at 60 °C. In contrast, the lithium grease exhibits a different trend. It maintains its viscosity to a higher extent as the temperature increases beyond 20 °C, indicating a higher viscosity index. This difference highlights the thermal behavior distinction between the two greases. Based on these findings, it is important to test the addition of nano-additives of higher thermal stability, such as ZnO, to enhance the thermal behavior of the palm grease sample.

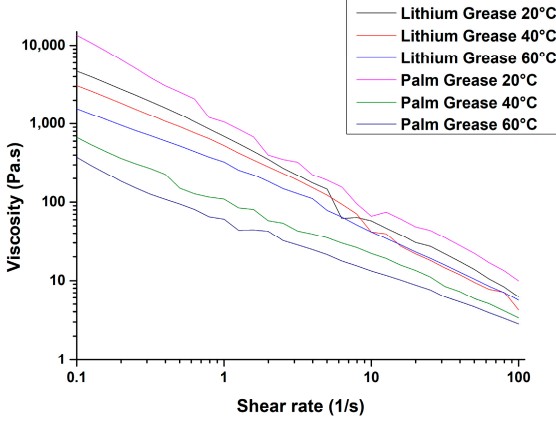

**Figure 7.** Rheological behavior of grease samples at different temperatures.

To confirm the rheological results, kinematic viscosity results of palm grease samples with and without nano-additives were determined. The kinematic viscosity values at 40 °C and at 100 °C of palm grease samples (without additives) were found to be 47.5 and 6.39 cSt, respectively, which are higher than those of lithium grease. This is due to the longer carbon chains in fatty acids (C18–C24) of large molecular weights which are responsible for creating a thicker film between the rubbing contacts with higher viscosity [81,82]. According to Table 2, the kinematic viscosity values have witnessed a remarkable improvement, especially when adding rGO, by percentage increase ranging from 22% to 36% as compared to palm oil without additives. The highest viscosity values were found in the base oil mixed with 0.1 wt.% ZnO and 2 wt.% rGO. However, the sample containing 2 wt.% rGO showed a slightly higher kinematic viscosity than that of sample containing 0.1 wt.% ZnO. ZnO particles may act as fillers or thickeners in the oil. When dispersed in the oil, it is believed that ZnO particles slightly increase the viscosity of the oil by impeding the flow and movement of oil molecules. On the other hand, rGO has a large surface area and can interact with oil molecules through van der Waals forces. These in-

teractions can promote the formation of clusters or agglomerates, resulting in increased intermolecular interactions within the oil. This increased cohesion among oil molecules can contribute to higher viscosity [49]. Hence, it reduces the shearing velocity of fluid layers, making it thicker and more resistant to flow. The high concentrations of rGO causes more coherency of nanoparticles due to their higher aggregation tendency than those of ZnO.

The pour point of palm grease reached 9 °C, while it is −15 °C for lithium grease. The presence of approximately 52% of saturated fatty acids in palm olein negatively affects the flow of lubricant at low temperatures, indicated by higher pour point values [18]. Fortunately, saturated fatty acids enhance the performance of palm oil under rough operating conditions in terms of better thermal oxidative stability [31]. rGO nano-additives at different concentrations experience no influence on the pour point, whereas the presence of ZnO reduced the pour point to 6 °C. This reduction in pour point is caused by the thermophysical characteristics of ZnO at lower temperatures allowing its nanoparticles to remain mobile while the temperature reaches 9 °C. Below 6 °C, the ZnO nanoparticles completely agglomerated due to van der Waal forces exhibiting no further enhancement in the pour point. This finding is proved by a previous investigation that tested ZnO and $MoS_2$ in diesel oil, showing a similar positive influence of ZnO on the pour point of diesel oil [49].

The oxidation stability results shown in Table 2 are obtained in terms of the time period till oxidation for samples of base oil with and without ZnO nano-additives. Palm oil is susceptible to structure oxidation, especially at high temperatures and pressures. It is primarily composed of palmitic acid (saturated fatty acid) and oleic acid (monounsaturated fatty acid), accounting for a total of 84.78% of the fatty acid profile [31,54]. The remaining small percentages of polyunsaturated fatty chains, which contains C=C bonds, are the main target for hydrolytic rancidity or oxidation leading to the formation of high molecular weight hydrocarbons, peroxide, and carbonyl compounds. During service time in machinery, this may lead to the formation of undesired carbonaceous sludge in oil. The presence of ZnO wt.% gradually improved the oxidation stability of base oil from 30 min to 48 min in the case of adding 0.5 wt.% ZnO with a total increase of 60%. Hence, ZnO delayed the degradation of the bio-oil properties due to its antioxidant nature [83]. The surface passivation provided by the ZnO-deposited layer on the metal surface restricts further oxidation of the Zn films, leading to improved oxidation and thermal stability [31,83]. ZnO also acts as a catalyst for the decomposition of hydroperoxides, which are formed during the initial stages of oxidation. By facilitating the decomposition of hydroperoxides, ZnO helps to prevent the propagation of oxidation reactions. Therefore, the addition of ZnO nano-additive provides more improvement of oxidation stability than other similar nanomaterials such as $TiO_2$, which was tested on base oils in previous work and achieved an increase of only 7% [46].

### 3.3. Tribological Properties Results

Figure 8a shows the results of load-carrying capacity obtained from the customized tribotest setup. The values of the maximum load before seizure takes place were recorded for each test grease sample. The lithium grease samples showed load-carrying capacities of 864 N, while palm grease without additives exhibited only 800 N.

The results also show that the addition of ZnO wt.% and rGO wt.% enhanced the load-carrying capacity of palm grease, where the improvement was shown to depend on the percentage added. For the lower concentrations of ZnO (0.05 wt.% and 0.1 wt.%), as well as for rGO (0.5 wt.%), the load-carrying capacity increased by around 30% compared to palm grease without additives. The grease sample containing 2 wt.% rGO nano-additive demonstrated an enhancement of 60% of the load-carrying capacity.

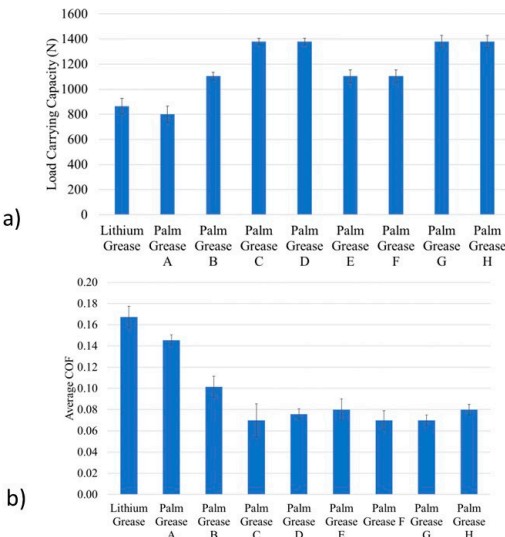

**Figure 8.** (**a**) Load-carrying capacity (nonseizure load) values. (**b**) Average friction coefficient values for test grease samples**.**

Figure 8b shows the determined COF values for tested grease samples. COF of palm grease without additives was found to be 0.1, which is lower than that of lithium grease by 15%. For samples (B) containing 0.05 wt.% ZnO nanoparticles, the COF was decreased by about 40%. Higher reduction of COF can reach 60% for sample (C) containing 0.1 wt.% ZnO. Similarly, specimens containing blends with 1 wt.% rGO nanosheets in palm grease showed a decrease by 60% of COF.

SEM was applied for the wear scar surface of each roller sample to visualize the worn surface morphology to provide more insight into the effect of each blend on the grease tribological behavior during the test. Figure 9a shows the wear scar surface of the roller sample lubricated with lithium grease. Wide, deep grooves with a rough surface morphology can be observed on the examined worn surface. By considering the obtained COF values for lithium grease, it can be concluded that the established tribolayer between contact surfaces during the tests provided inadequate protection against friction. Increasing the applied weights leads to a gradual failure of the lithium grease protective film, which in turn, causes a rapid increase in COF value and associated severe rubbing action in surface morphology.

In Figure 9b, the worn surface lubricated with palm grease is shallower, with small irregular embossments from rubbing action. Palm oil fatty acids caused a reduction in the damage to the worn surface along with COF enhancement due to their long chains along with the existing polar groups, which facilitate the strong interaction with the rubbing contact [31]. For blends with nano-additives (Figure 9c–i), the furrows become superficial and mild and the worn surface images show a smooth topography with narrow wear markings.

For samples with rGO concentrations (Figure 9g–i), the graphene 2D nanoparticles form strong absorption layers with a high elastic modulus, which contributes to the observed increase in load-carrying capacity and reduces the damage of the worn surface. Also, the self-lubricating characteristic of the carbonaceous material greatly reduces the COF in comparison with lithium and palm grease without additives. ZnO has the ability to chemically react with the surface layer of the metal, forming zinc compounds which adhere to the surface, forming a protective boundary lubricating film. This formed layer is well known to possess a low shear strength which acts as a lubricating film to prevent surface asperities from coming into contact and reduces friction between moving parts.

However, the increase in COF and corresponding morphology of wear scar surface at higher concentrations of ZnO and rGO is attributed to the ability of the ZnO and rGO

nanoparticles to reduce the oil flow, leading to a significant increase in the palm oil viscosity. Another possible reason is the ability of ZnO and rGO at larger concentrations, in some cases, to collide and agglomerate, leading to larger particles and causing more wear and COF, especially in the absence of dispersive agents [84].

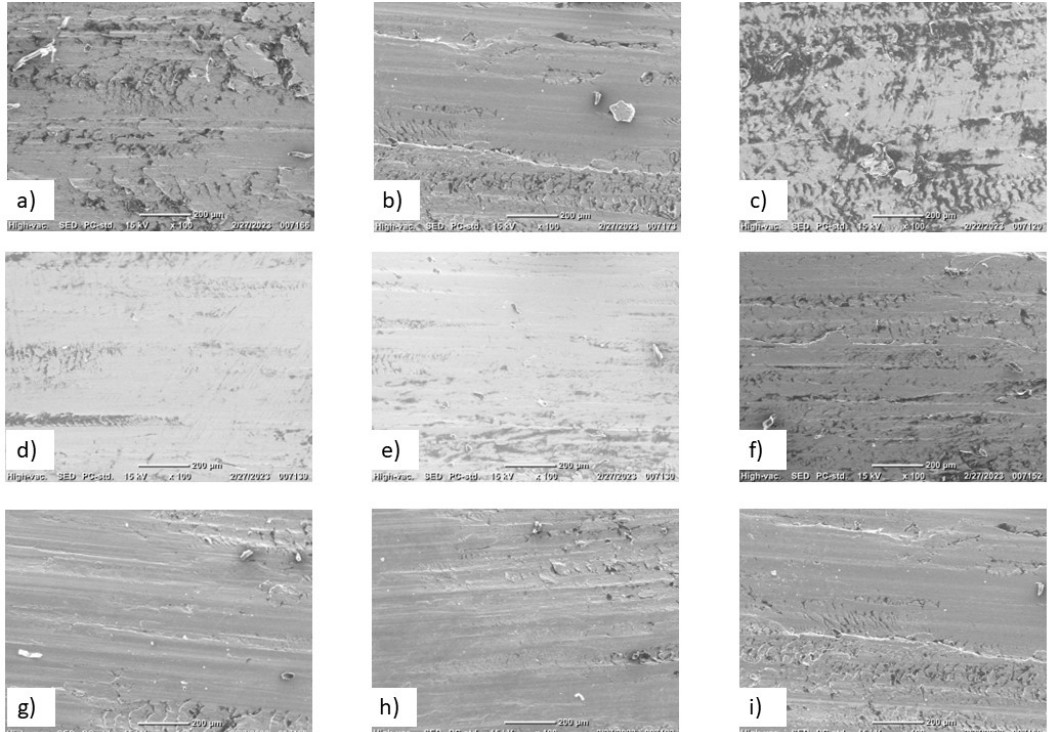

**Figure 9.** SEM images at magnification 200× of the wear scars from the customized tribotester setup for test samples: (**a**) lithium grease; (**b**) palm grease; (**c**) 0.05 wt.% ZnO; (**d**) 0.1 wt.% ZnO; (**e**) 0.25 wt.% ZnO; (**f**) 0.5 wt.% ZnO; (**g**) 0.5 wt.% rGO; (**h**) 1 wt.% rGO; (**i**) 2 wt.% rGO.

## 4. Conclusions

In this work, palm grease was synthesized and tested for chemical–physical and tribological properties for industrial applications. ZnO and rGO were selected as nano-additives to be blended with synthesized palm grease samples at different concentrations. Test results led to the following conclusions:

- Penetration test results of synthesized palm grease without additives showed a rigidity corresponding to NLGI grade 3; therefore, it is suitable for the lubrication of rotating machinery components. The roll stability test confirmed the penetration test results, ensuring the superiority of palm grease over lithium grease. The existence of stearic acids in palm oil is responsible for creating closed grease structures with long entangled fibers, enhancing its stability.
- The copper corrosion test showed that palm grease with and without additives indicates superb protection of mating surfaces against corrosion and absence of harmful sulfur in palm oil.
- Palm grease showed higher dropping point (209 °C) than that of lithium grease (160 °C), which indicates better thermal stability and coherency.
- Kinematic viscosity values at 100 °C of palm grease samples were higher than those of lithium grease by 40%. The addition of ZnO and rGO nano-additives at different weight percentages enhanced the kinematic viscosity at 100 °C up to 25%.
- ZnO is shown to play a major role in reducing the oxidation tendency of palm grease with a total increase in oxidation time of 60%. It also reduced the pour point of palm grease from 9 °C to 6 °C, which enables palm grease to be used at lower-temperature conditions.

- Adding rGO and ZnO at increasing ratios significantly enhanced the load-carrying capacity up to 60%, as compared to palm grease without additives. The addition of ZnO and rGO contributed to reduction in the COF from 0.1 to 0.07.

For future work, hybrid mixtures of ZnO and rGO with suitable dispersing agents are to be tested for COF and EP properties. Also, the grease samples with the tested blends can be applied to rolling element bearings and gears.

**Author Contributions:** Conceptualization, G.A.N., M.G.A.N., B.G.N. and M.E.; methodology, M.G.A.N., B.G.N. and H.S.H.; samples preparation, B.G.N., H.S.H. and M.G.A.N.; validation, M.G.A.N., B.G.N. and G.A.N.; formal analysis, M.G.A.N., B.G.N. and F.P.; investigation, M.G.A.N., M.E. and B.G.N.; resources, M.E., F.P. and M.G.A.N.; data curation, M.G.A.N., B.G.N., H.S.H. and M.E.; writing—original draft preparation, B.G.N., F.P. and M.G.A.N.; writing—review and editing, G.A.N., F.P. and H.S.H.; visualization, M.G.A.N. and M.E.; project administration, G.A.N. All authors have read and agreed to the published version of the manuscript.

**Funding:** This research received no external funding.

**Data Availability Statement:** Data available on request due to privacy restrictions. The data presented in this study are available on request from the corresponding author. The data are not publicly available due to being filed as a patent.

**Acknowledgments:** We extend our heartfelt thanks to Netzsch GmbH in Germany for their invaluable support with the rotational rheometer. Furthermore, we are deeply grateful to the Institute for Multiphase Processes (IMP) at Leibniz University Hannover, Germany, for their indispensable assistance in conducting FTIR tests for base oil and grease samples. We also wish to acknowledge the Faculty of Science at Alexandria University, Egypt, and the City of Scientific Research and Technological Applications for their generous contributions to this study in nano-additive characterization.

**Conflicts of Interest:** The authors declare no conflicts of interest.

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
