# Peer review of "Tribological and Chemical–Physical Behavior of a Novel Palm Grease Blended with Zinc Oxide and Reduced Graphene Oxide Nano-Additives"

_lubricants, doi:10.3390/lubricants12060191_

Round 1

Reviewer 1 Report

Comments and Suggestions for Authors 1. Some theoretical explanations can be added appropriately in the abstract section.
2.The introduction section can be appropriately simplified.
3. In the synthesisi part, Afterward, the temperature is increased to 250℃ with 208 simultaneous mixing for 1 hr. in order to ensure the soap dissolution. , A reaction at 250℃ may undergo oxidation, usually around 220℃.
4.An XPS spectrum of the worn steel ball surface can be added to explore the chemical valence states of GO and Zn after tribological process. Comments on the Quality of English Language

Minor editing of English language required

Author Response

Dear Respectable Reviewer,

On behalf of the authors, I would like to express my deepest gratitude for your valuable comments and the careful checking of the paper. We believe that the modifications that we have made based on your comments have improved the quality of the article.  Kindly find below the detailed responses to your comments.

Reply to Reviewer# 1 comments:

  1. Some theoretical explanations can be added appropriately in the abstract section.

Response: Thank you very much for this valuable comment. We have updated the abstract accordingly while considering the size limit of the abstract section according to the journal guidelines.

  1. The introduction section can be appropriately simplified.

Response: Thank you very much for this essential remark. We have considered this by reducing the size of the introduction section in different appropriate locations.

  1. In the synthesisi part, “Afterward, the temperature is increased to 250℃with 208 simultaneous mixing for 1 hr. in order to ensure the soap dissolution. ”, A reaction at 250℃ may undergo oxidation, usually around 220℃.

Response: We appreciate your important note. Your concern is indeed valid, as oxidation can occur at high temperatures. However, during the heating process, we took several precautions to prevent oxidation. The flask was appropriately covered to avoid direct contact with atmospheric oxygen. Additionally, we ensured that the heating process was conducted gradually and well-controlled to avoid localized overheating. Along with all of these precautions, the FTIR of the palm grease (after synthesis) compared to palm oil (before synthesis) assures that there was no obvious change in the spectrum except for the carboxylate ion (COO–) group for the lithium hydroxystearate and the increased intensity of C-H stretching band.

  1. An XPS spectrum of the worn steel ball surface can be added to explore the chemical valence states of GO and Zn after tribological process.

Response: Thank you for your valuable comment. Unfortunately, due to the unavailability of XPS at our institution, we are unable to utilize this technique. Our study primarily focuses on comparing the effects of ZnO and rGO to evaluate their efficiency in reducing friction and wear. We appreciate your understanding and hope that the comparative analysis we have provided will still offer meaningful insights.

Reviewer 2 Report

Comments and Suggestions for Authors

Dear authors,

Your manuscript is an interesting work in the direction to obtain biological lubricants in this world context.

Some minor comments:

1. You presented 8 sorts of Palm Greases with various  percent of nano-aditives.  In the Table 2 are not indicated the most important properties for all greases! All the eight sort of greases are compared with a commercial Lithium Grease. In Conclusion is afirmed that the penetration NLG grade is 1 but in table 2 arenot indicated.

Also, the Oxidation stability for the commercial Lithium Grease is not indicated to can realize a good comparation. 

In my opinion for a Lithium grease the oxidation Stability is between 150-250 minutes. 

2. The tribological tests are realized with a cylinder on disc tribometer. Have you realized tests with Four Ball Machine?

3. Please indicate the forces for determining of the COF!

Author Response

Dear Respectable Reviewer,

On behalf of the authors, I would like to express my deepest gratitude for your valuable comments and the careful checking of the paper. We believe that the modifications that we have made based on your comments have improved the quality of the article.

Kindly find below the detailed responses to your comments and an information on the modifications made.

Reply to Reviewer# 2 comments:

  1. You presented 8 sorts of Palm Greases with various percent of nano-aditives.  In the Table 2 are not indicated the most important properties for all greases! All the eight sort of greases are compared with a commercial Lithium Grease. In Conclusion is afirmed that the penetration NLG grade is 1 but in table 2 arenot indicated.

Response: We appreciate your keen observation in this issue. We agree with you that in table 2 we have presented the main properties of palm grease and palm oil without additives against commercial lithium grease and mineral oil. Regarding the other palm grease samples with different blends of additives, we would like to emphasize on the fact that our paper’s objective was to study the influence of rGO and ZnO on certain properties that represented challenge of applying palm oil in industrial lubricant mentioned in the abstract section. These properties are kinematic viscosity at 100 degrees, copper corrosion, oxidation stability, pour point beside tribological properties of palm oil and grease (which are described in other section).

As for the NLGI grade, we are sorry for the type mistake in the conclusions part. The obtained NLGI for palm grease is grade 3 unlike the lithium grease which is grade 2. We have corrected this typo and updated table 2 in the unworked penetration values to include the NLGI grade for both grease types. Thank you again.

  1. Also, the Oxidation stability for the commercial Lithium Grease is not indicated to can realize a good comparation. In my opinion for a Lithium grease the oxidation Stability is between 150-250 minutes.

Response: Again, thank you for the helpful remark. We agree that in the review of literature, commercial lithium grease has higher oxidation stabilities in terms of minutes according to the ASTM D8206 for greases. However, we have conducted the oxidation stability test on base oils only not the grease because we were more concerned with the response of the base oil itself and also we were constrained with the available equipment that follows the standard ASTM D 2272 for steam turbine oils as harsh test. The standard is different from the ASTM D 8206–18 for lubricating greases. We have also used this standard which also supported by reference [68] which tested oxidation stability on similar base oil for commercial greases. The results of the base oil in the reference are found around 27 min for the tested commercial mineral oils. We have added the values in ref [68] as guided values in the revised manuscript in table 2.

  1. The tribological tests are realized with a cylinder on disc tribometer. Have you realized tests with Four Ball Machine?

Response: Thank you very much for this remark. We have already conducted the tribological tests using customized test setup inspired by roller on ring Brugger test machine and also the Four Ball Machine. The information about the test can be found in the experimental work section in the lines 298 to 326. Also, figure (2) shows the principle of both the customized test and the four ball wear test.

  1. Please indicate the forces for determining of the COF!

Response: Thank you for your comment. We have followed the standard ASTM D 5183 in which the upper ball inside the cup is pressed against three lower balls using a static force of 40 kgf for 60 min ± 1 min then an incremental weight of 10 kg is added every 10 minutes until reaching the seizure condition. The software of the available 4-ball wear test equipment provides us with the coefficient of friction value by the end of test, i.e. the seizure point. We have included this force value and procedure of test in detail in the manuscript in section 2.11.

Reviewer 3 Report

Comments and Suggestions for Authors

The manuscript is well formatted, sound and easy to follow. It includes all necessary ASTM standards and is recommended to publishing after several clarifications. I have some minor questions/suggestions to it:

1. For most experiments three tests were performed to get an average, however, I don't find it sufficient. A good rule of thumb is to run five tests, as it provided better repeatability gauge. Also the error margins are stated only in fig. 8, while it'd be helpful to have them added to all values to understand the tests accuracy.

2. In table 2 most of the values are missing for the samples with nano additives. Could you provide the explanation?

3. Could you please explain how dis you determine the 'four less folded and stacked layers' (389) by TEM? I just don't see it and would love to get some guidelines where to look.

4. Another question is about 80:20 oil:thickener ratio choice, was it dictated by the commercial grease? Did you tested any other concentrations to see a performance difference?

5. Why 232C temperature was chosen in the drop point test? As well as 20, 40 and 60C for rheology?

6. I feel like the concentration of ZnO or CO is missing in line 564: 'For samples containing 0.05 ZnO nanoparticles, the COF is decreased by about 40%. Higher reduction of COF can reach (60 %) for sample containing. Similarly, specimens containing blends with 1 wt.% rGO nanosheets in palm grease showed a decrease by 60% of COF'. As well as 'wt.%' in front of 0.05 ZnO.

7. It would be beneficial to add a few sentences at the end of Introduction section highlighting the uniqueness of this work in comparison to all provided literature background.

8. I noticed that you stated that will use the mixtures of nano additives in the future work, my question is why haven't you tried a least some, especially, those most promising individual compositions to possibly achieve a compounding effect.

Comments on the Quality of English Language

1. Just a few super- and subscript edits throughout the text: 9 oC to 6 oC (35), CO2 (46), TiO2 (129, 138, 545), CH2 (362), C-H2 (380).

2. There are a few repetitions which could be reworded, for example word 'improve' in lines 552-556.

Author Response

Dear Respectable Reviewer,

On behalf of the authors, I would like to express my deepest gratitude for your valuable comments and the careful checking of the paper. We believe that the modifications that we have made based on your comments have improved the quality of the article. Kindly find below the detailed responses to your comments.

Reply to Reviewer# 3 comments:

  1. For most experiments three tests were performed to get an average, however, I don't find it sufficient. A good rule of thumb is to run five tests, as it provided better repeatability gauge. Also the error margins are stated only in fig. 8, while it'd be helpful to have them added to all values to understand the tests accuracy.

Response: Thank you so much for your valuable remark. We have tried to conduct each experiment for physical, chemical, and tribological properties with sufficient repeated number of times to make sure that we receive independent results obtained by same operators working on almost identical test samples. We agree with you that five tests would provide more accurate representation of results. However, and due to the large number of tests conducted in this work with large amounts of prepared samples, we have selected three times of repetitions for the tests in this manuscript that achieved acceptable standard deviations among results from our point of view. We have added the error margins in table 2 for the tests in which we recorded noticable standard deviation. We also wish for your understanding in this regard and we will consider this important note in the future work.

  1. In table 2 most of the values are missing for the samples with nano additives. Could you provide the explanation?

Response:  We appreciate your keen observation in this issue. We agree with you that in table 2 some properties are not calculated for samples with nano-additives. First, we have presented the main properties of palm grease and palm oil without additives against commercial lithium grease and mineral oil. Regarding the other palm grease samples with different blends of additives, we would like to emphasize on the fact that our paper’s objective was to study the influence of rGO and ZnO on certain properties that represented challenge of applying palm oil in industrial lubricant mentioned in the abstract section. These properties are kinematic viscosity at 100 degrees, copper corrosion, oxidation stability, pour point beside tribological properties of palm oil and grease (which are described in other section). We have also included in the revised manuscript in table 2 the NLGI grade for both grease types and the oxidation stability value for mineral base oil.

  1. Could you please explain how dis you determine the 'four less folded and stacked layers' (389) by TEM? I just don't see it and would love to get some guidelines where to look.

Response: Thank you for your insightful question. The determination of the "four less folded and stacked layers" by TEM was achieved through multiple methods. Firstly, the transparent nature of graphene implies a low number of layers, suggesting the presence of few-layer graphene. Secondly, upon observation of TEM images, we identified approximately four separated edges in the thick edge of the platelet, further indicating the presence of a small number of layers. Additionally, the TEM findings were supported by Raman spectroscopy results, where the I2D/IG intensity ratio was measured at 0.91, characteristic of few-layer graphene (lines 393 - 402). This combined approach, utilizing both TEM and Raman spectroscopy, provided a comprehensive understanding of the graphene's layer structure.

  1. Another question is about 80:20 oil:thickener ratio choice, was it dictated by the commercial grease? Did you tested any other concentrations to see a performance difference?

Response: Thank you for your valuable comment. We conducted experiments with various percentages of thickener, including both lower and higher concentrations than 20%. However, increasing the thickener concentration beyond 20% resulted in excessively high firmness and agglomeration within the mixture. Conversely, decreasing the concentration below 20% yielded a grease sample that was too soft to meet our criteria. Therefore, after careful consideration of these factors, we opted to rely on a 20% thickener percentage for our formulation.

  1. Why 232C temperature was chosen in the drop point test? As well as 20, 40 and 60C for rheology?

Response: Thank you for this essential remark. The dropping point of the grease samples was measured in accordance with ASTM D2265. The standard provides a range of specific temperatures to test at, and after careful consideration, we selected 232℃ (the second recommended temperature according to the standard). This temperature was chosen with the expectation that the drop point would not deviate significantly from that of commercial grease. Additionally, we attempted testing at 121℃ (the first recommended temperature according to the standard), but no notable change occurred in the samples.

As for the rheology, we increased the temperatures to observe the effect on the viscosity index and the thermal stability of the grease. This gradual increase was selected to systematically record the rheological behavior at different temperatures, providing a comprehensive understanding of how the grease performs under varying thermal conditions.

  1. I feel like the concentration of ZnO or CO is missing in line 564: 'For samples containing 0.05 ZnO nanoparticles, the COF is decreased by about 40%. Higher reduction of COF can reach (60 %) for sample containing. Similarly, specimens containing blends with 1 wt.% rGO nanosheets in palm grease showed a decrease by 60% of COF'. As well as 'wt.%' in front of 0.05 ZnO.

Response: We are very sorry for these missing parts. we have gone through these parts and completed the missing information in in the revised manuscript.

  1. It would be beneficial to add a few sentences at the end of Introduction section highlighting the uniqueness of this work in comparison to all provided literature background.

Response: Thank you for this valuable comment. We have considered this in the revised manuscript.

  1. I noticed that you stated that will use the mixtures of nano additives in the future work, my question is why haven't you tried a least some, especially, those most promising individual compositions to possibly achieve a compounding effect.

Response: Thank you for your question. In response, we have chosen to address the synergy between the nanoadditives in a separate manuscript. Our main objective in this study was to synthesize a bio-grease that can compete with commercial alternatives. Investigating the synergistic effects of nanoadditives on the properties of bio-grease requires a dedicated study, as it involves carefully selecting the appropriate percentages of the combined nanoadditives to prevent chemical reactions and agglomeration within the grease structure. Therefore, in this phase of work we wanted to understand the individual effect of each nanoadditives on the properties of the grease before moving forward to combine them.

Comments on the Quality of English Language

  1. Just a few super- and subscript edits throughout the text: 9 oC to 6 oC (35), CO2 (46), TiO2 (129, 138, 545), CH2 (362), C-H2 (380).
  2. There are a few repetitions which could be reworded, for example word 'improve' in lines 552-556.

Response: Thank you very much for these valuable marks. We are sorry for these typos and we have modified these lines accordingly.
